# Scaling Neural Tangent Kernels
# via Sketching and Random Features

**Amir Zandieh**[*]
Max-Planck-Institut für Informatik
azandieh@mpi-inf.mpg.de

**Insu Han**[*]
Yale University
insu.han@yale.edu

**Haim Avron**
Tel Aviv University
haimav@tauex.tau.ac.il

**Neta Shoham**
Tel Aviv University
shohamne@gmail.com

**Chaewon Kim**
KAIST
chaewonk@kaist.ac.kr

**Jinwoo Shin**
KAIST
jinwoos@kaist.ac.kr

## Abstract

The Neural Tangent Kernel (NTK) characterizes the behavior of infinitely-wide neural networks trained under least squares loss by gradient descent. Recent works also report that NTK regression can outperform finitely-wide neural networks trained on small-scale datasets. However, the computational complexity of kernel methods has limited its use in large-scale learning tasks. To accelerate learning with NTK, we design a near input-sparsity time approximation algorithm for NTK, by sketching the polynomial expansions of arc-cosine kernels: our sketch for the convolutional counterpart of NTK (CNTK) can transform any image using a linear runtime in the number of pixels. Furthermore, we prove a spectral approximation guarantee for the NTK matrix, by combining random features (based on leverage score sampling) of the arc-cosine kernels with a sketching algorithm. We benchmark our methods on various large-scale regression and classification tasks and show that a linear regressor trained on our CNTK features matches the accuracy of exact CNTK on CIFAR-10 dataset while achieving $150\times$ speedup.

## 1 Introduction

Recent results have shown that over-parameterized Deep Neural Networks (DNNs), generalize surprisingly well. In an effort to understand this phenomena, researchers have studied ultra-wide DNNs and shown that in the infinite width limit, a fully connected DNN trained by gradient descent under least-squares loss is equivalent to kernel regression with respect to the Neural Tangent Kernel (NTK) [5, 11, 22, 28]. This connection has shed light on DNNs' ability to generalize [10, 34] and optimize (train) their parameters efficiently [3, 4, 16]. More recently, Arora et al. [5] proved an analogous equivalence between convolutional DNNs with infinite number of channels and Convolutional NTK (CNTK). Beyond the aforementioned theoretical purposes, several papers have explored the algorithmic use of this kernel. Arora et al. [6] and Geifman et al. [19] showed that NTK based kernel models can outperform trained DNNs (of finite width). Additionally, CNTK kernel regression sets an impressive performance record on CIFAR-10 for kernel methods without trainable kernels [5]. The NTK has also been used in experimental design [39] and predicting training time [43].

However, the NTK-based approaches encounter the computational bottlenecks of kernel learning. In particular, for a dataset of $n$ images $x_1, x_2, \ldots x_n \in \mathbb{R}^{d \times d}$, only writing down the CNTK kernel matrix requires $\Omega\left(d^4 \cdot n^2\right)$ operations [5]. Running regression or PCA on the resulting kernel matrix takes additional cubic time in $n$, which is infeasible in large-scale setups.

---

[*]Equal contribution.

35th Conference on Neural Information Processing Systems (NeurIPS 2021).

There is a rich literature on kernel approximations for large-scale learning. One of the most popular approaches is the *random features method* which works by randomly sampling the feature space of the kernel function, originally due to the seminal work of Rahimi and Recht [37]. Another popular approach which is developed in linear sketching literature [41], works by designing sketches that can be efficiently applied to the feature space of a kernel without needing to explicitly form the high dimensional feature space. This approach has been successful at designing efficient subspace embeddings for the polynomial kernel [7, 1]. In this paper, we propose solutions for scaling the NTK and CNTK by building on both of these kernel approximations techniques and designing efficient feature maps that approximate the NTK/CNTK evaluation. Consequently, we can simply transform the input dataset to these feature spaces, and then apply fast linear learning methods to approximate the answer of the corresponding nonlinear kernel method efficiently. The performance of such approximate methods is similar or sometimes better than the exact kernel methods due to implicit regularization effects of the approximation algorithms [37, 38, 23].

## 1.1 Overview of Our Contributions

• One of our results is an efficient random features construction for the NTK. Our starting point is the explicit NTK feature map suggested by Bietti and Mairal [9] based on tensor product of the feature maps of arc-cosine kernels. We obtain our random features, by sampling the feature space of arc-cosine kernels [12]. However, the naïve construction of the features would incur an exponential cost in the depth of the NTK, due to the tensor product of features generated in consecutive layers. We remedy this issue, by utilizing an efficient sketching algorithm for tensor products known as TENSORSRHT [1] which can effectively approximate the tensor products of vectors while preserving their inner products. We provide a rigorous error analysis of the proposed scheme in Theorem 2.

• Our next results are sketching methods for both NTK and CNTK using a runtime that is linearly proportional to the sparsity of the input dataset (or number of pixels of images). Our methods rely on the arc-cosine kernels' feature space defined by their Taylor expansion. By careful truncation of their Taylor series, we approximate the arc-cosine kernels with bounded-degree polynomial kernels. Because the feature space of a polynomial kernel is the tensor product of its input space, its dimensionality is exponential in the degree of the kernel. Fortunately, Ahle et al. [1] have developed a linear sketch known as POLYSKETCH that can reduce the dimensionality of high-degree tensor products very efficiently, therefore, we can sketch the resulting polynomial kernels using this technique. We then combine the transformed features from consecutive layers by further sketching their tensor products. In case of CNTK, we have an extra operation which sketches the direct sum of the features of neighbouring pixels at each layer that precisely corresponds to the convolution operation in CNNs. We carefully analyze the errors introduced by polynomial approximations and various sketching steps in our algorithms and also bound their runtimes in Theorems 1 and 4.

• Furthermore, we improve the arc-cosine random features to spectrally approximate the entire kernel matrix, which is advocated in recent literature for ensuring high approximation quality in downstream tasks [8, 32]. Our construction is based on leverage score sampling, which entertains better convergence bounds [8, 28, 29]. However, computing this distribution is as expensive as solving the kernel methods exactly. We propose a simple distribution that tightly upper bounds the leverage scores of arc-cosine kernels and for further efficiency, use Gibbs sampling to generate random features from our proposed distribution. We provide our spectral approximation guarantee in Theorem 3.

• Finally, we empirically benchmark our proposed methods on various classification/regression tasks and demonstrate that our methods perform similar to or better than exact kernel method with NTK and CNTK while running extremely faster. In particular, we classify CIFAR-10 dataset $150\times$ faster than exact CNTK and at the same time achieve higher test accuracy.

## 1.2 Related Works

There has been a long line of work on the correspondence between DNN and kernel machines [26, 30, 35, 18, 42]. Furthermore, there has been many efforts in understanding a variety of NTK properties including optimization [27, 3, 16, 44], generalization [10], loss surface [31], etc.

Novak et al. [35] tried accelerating CNTK computations via Monte Carlo methods by taking the gradient of a randomly initialized CNN with respect to its weights. Although they do not theoretically bound the number of required features, the fully-connected version of this method is analyzed in [5].

Particularly, for the gradient features to approximate the NTK up to $\varepsilon$, the network width needs to be $\Omega(\frac{L^6}{\varepsilon^4} \log \frac{L}{\delta})$, thus, transforming a single vector $x \in \mathbb{R}^d$ requires $\Omega(\frac{L^{13}}{\varepsilon^8} \log^2 \frac{L}{\delta} + \frac{L^6}{\varepsilon^4} \log \frac{L}{\delta} \cdot \mathrm{nnz}(x))$ operations, which is slower than our Theorem 1 by a factor of $L^3/\varepsilon^2$. Furthermore, [5] shows that the performance of these random gradients is worse than exact CNTK by a large margin, in practice. More recently, [28] proposed leverage score sampling for the NTK, however, their work is primarily theoretical and suggests no practical way of sampling the features. Another line of work on NTK approximation is an explicit feature map construction via tensor product proposed by Bietti and Mairal [9]. These explicit features can have infinite dimension in general and even if one uses a finite-dimensional approximation to the features, the computational gain of random features will be lost due to expensive tensor product operations.

A popular line of work on kernel approximation problem is based on the Fourier features method [37], which works well for shift-invariant kernels and with some modifications can embed the Gaussian kernel near optimally [8]. Other random feature constructions have been suggested for a variety of kernels, e.g., arc-cosine kernels [12], polynomial kernels [36]. In linear sketching literature, Avron et al. [7] proposed a subspace embedding for the polynomial kernel which was recently extended to general dot product kernels [20]. The runtime of this method, while nearly linear in sparsity of the input dataset, scales exponentially in kernel's degree. Recently, Ahle et al. [1] improved this exponential dependence to polynomial which enabled them to sketch high-degree polynomial kernels and led to near-optimal embeddings for Gaussian kernel. In fact, this sketching technology constitutes one of the main ingredients of our proposed methods. Additionally, combining sketching with leverage score sampling can improve the runtime of the polynomial kernel embeddings [40].

## 1.3 Preliminaries: POLYSKETCH and TENSORSRHT Transforms

**Notations.** We use $[n] := \{1, \ldots, n\}$. We denote the tensor (a.k.a. Kronecker) product by $\otimes$ and the element-wise (a.k.a. Hadamard) product of two vectors or matrices by $\odot$. Although tensor products are multidimensional objects, we often associate $x \otimes y$ with a single dimensional vector $(x_1 y_1, x_2 y_1, \ldots x_m y_1, x_1 y_2, \ldots x_m y_2, \ldots x_m y_n)$. For shorthand, we use the notation $x^{\otimes p}$ to denote $\underbrace{x \otimes \ldots \otimes x}_{p \text{ terms}}$, the $p$-fold self-tensoring of $x$. Another related operation that we use is the *direct sum* of vectors: $x \oplus y := \begin{bmatrix} x^\top, y^\top \end{bmatrix}^\top$. We need notation for *sub-tensors* of a tensor. For instance, for a 3-dimensional tensor $\boldsymbol{Y} \in \mathbb{R}^{m \times n \times d}$ and every $l \in [d]$, we denote by $\boldsymbol{Y}_{(:,:,l)}$ the $m \times n$ matrix that is defined as $\left[ \boldsymbol{Y}_{(:,:,l)} \right]_{i,j} := \boldsymbol{Y}_{i,j,l}$ for $i \in [m], j \in [n]$. For square matrices $\boldsymbol{A}$ and $\boldsymbol{B}$, we write $\boldsymbol{A} \preceq \boldsymbol{B}$ if $\boldsymbol{B} - \boldsymbol{A}$ is positive semi-definite. We also denote $\mathrm{ReLU}(x) = \max(x, 0)$ and consider this element-wise operation when the input is a matrix. We use $\mathrm{nnz}(x)$ to denote the number of nonzero entries in $x$. Given a positive semidefinite matrix $\boldsymbol{K}$ and $\lambda > 0$, the statistical dimension of $\boldsymbol{K}$ with $\lambda$ is defined as $s_\lambda(\boldsymbol{K}) := \mathrm{tr}(\boldsymbol{K}(\boldsymbol{K} + \lambda \boldsymbol{I})^{-1})$. For two functions $f$ and $g$ we denote their twofold composition by $f \circ g$, defined as $f \circ g(\alpha) := f(g(\alpha))$.

The TENSORSRHT is a norm-preserving dimensionality reduction that can be applied to the tensor product of two vectors very quickly [1]. This transformation is a generalization of the Subsampled Randomized Hadamard Transform (SRHT) [2] and can be computed in near linear time using the FFT algorithm. The POLYSKETCH extends the idea behind TENSORSRHT to high-degree tensor products by recursively sketching pairs of vectors in a binary tree structure. This sketch preserves the norm of vectors in $\mathbb{R}^{d^p}$ with high probability and can be applied to tensor product vectors very quickly. The following Lemma, summarizes Theorems 1.2 and 1.3 of [1] and is proved in Appendix B.

**Lemma 1** (POLYSKETCH)**.** *For every integers $p, d \geq 1$ and every $\varepsilon, \delta > 0$, there exists a distribution on random matrices $\boldsymbol{Q}^p \in \mathbb{R}^{m \times d^p}$, called degree $p$ POLYSKETCH such that **(1)** for some $m = \mathcal{O}\left( \frac{p}{\varepsilon^2} \log^3 \frac{1}{\varepsilon \delta} \right)$ and any $y \in \mathbb{R}^{d^p}$, $\Pr\left[ \|\boldsymbol{Q}^p y\|_2^2 \in (1 \pm \varepsilon) \|y\|_2^2 \right] \geq 1 - \delta$; **(2)** for any $x \in \mathbb{R}^d$, if $e_1 \in \mathbb{R}^d$ is the standard basis vector along the first coordinate, the total time to compute $\boldsymbol{Q}^p(x^{\otimes(p-j)} \otimes e_1^{\otimes j})$ for all $j = 0, 1, \ldots, p$ is $\mathcal{O}\left( pm \log^2 m + \min\left\{ \frac{p^{3/2}}{\varepsilon} \log \frac{1}{\delta} \mathrm{nnz}(x), pd \log d \right\} \right)$; **(3)** for any collection of vectors $v_1, \ldots, v_p \in \mathbb{R}^d$, the time to compute $\boldsymbol{Q}^p(v_1 \otimes \cdots \otimes v_p)$ is bounded by $\mathcal{O}\left( pm \log m + \frac{p^{3/2}}{\varepsilon} d \log \frac{1}{\delta} \right)$; **(4)** for any $\lambda > 0$ and any matrix $\boldsymbol{A} \in \mathbb{R}^{d^p \times n}$, where the statis-*

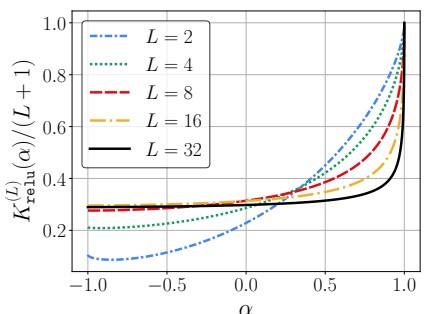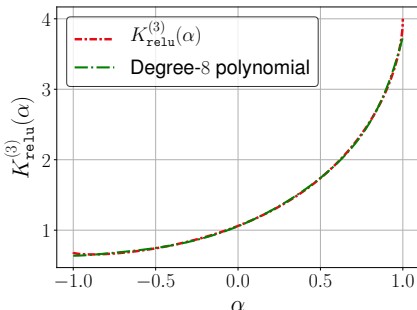

Figure 1: (Left) Normalized ReLU-NTK function $K_{\text{relu}}^{(L)}(\cdot)$ for $L = \{2, 4, 8, 16, 32\}$ and (Right) a degree-8 polynomial approximation of ReLU-NTK with $L = 3$.

*tical dimension of $\boldsymbol{A}^\top \boldsymbol{A}$ is $s_\lambda$, there exists some $m = \mathcal{O}\left(\frac{p^4 s_\lambda}{\varepsilon^2} \log^3 \frac{n}{\varepsilon\delta}\right)$ such that,*

$$\Pr\left[(1 - \varepsilon)\left(\boldsymbol{A}^\top \boldsymbol{A} + \lambda \boldsymbol{I}\right) \preceq (\boldsymbol{Q}^p \boldsymbol{A})^\top (\boldsymbol{Q}^p \boldsymbol{A}) + \lambda \boldsymbol{I} \preceq (1 + \varepsilon)\left(\boldsymbol{A}^\top \boldsymbol{A} + \lambda \boldsymbol{I}\right)\right] \geq 1 - \delta. \quad (1)$$

## 2 ReLU Neural Tangent Kernel

Arora et al. [5] showed how to exactly compute the NTK of a $L$-layer fully-connected network, denoted by $\Theta_{\text{ntk}}^{(L)}(y, z)$, for any pair of vectors $y, z \in \mathbb{R}^d$ using a dynamic program in $\mathcal{O}(d + L)$ time. However, it is hard to gain insight into the structure of this kernel using that the dynamic program expression which involves recursive applications of nontrivial expectations. Fortunately, for the important case of ReLU activation this kernel takes an extremely nice and highly structured form. The NTK in this case can be fully characterized by a univariate function $K_{\text{relu}}^{(L)} : [-1, 1] \to \mathbb{R}$ that we refer to as *ReLU-NTK*, which is the composition of the arc-cosine kernels [12] and was recently derived in [9]. Exploiting this special structure is the key to designing efficient sketching methods and random features for this kernel.

**Definition 1** (ReLU-NTK function). For every integer $L > 0$, the $L$-layer ReLU-NTK function $K_{\text{relu}}^{(L)} : [-1, 1] \to \mathbb{R}$ is defined via following procedure, for every $\alpha \in [-1, 1]$:

1. Let $\kappa_0(\alpha)$ and $\kappa_1(\alpha)$ be $0^{th}$ and $1^{st}$ order arc-cosine kernels [12] defined as follows,

$$\kappa_0(\alpha) := \frac{1}{\pi}\left(\pi - \arccos(\alpha)\right), \quad \text{and} \quad \kappa_1(\alpha) := \frac{1}{\pi}\left(\sqrt{1 - \alpha^2} + \alpha \cdot (\pi - \arccos(\alpha))\right). \quad (2)$$

2. Let $\Sigma_{\text{relu}}^{(0)}(\alpha) := \alpha$ and for $\ell = 1, 2, \ldots L$, define $\Sigma_{\text{relu}}^{(\ell)}(\alpha)$ and $\dot{\Sigma}_{\text{relu}}^{(\ell)}(\alpha)$ as follows,

$$\Sigma_{\text{relu}}^{(\ell)}(\alpha) := \underbrace{\kappa_1 \circ \kappa_1 \circ \cdots \circ \kappa_1}_{\ell\text{-fold self composition}}(\alpha), \quad \text{and} \quad \dot{\Sigma}_{\text{relu}}^{(\ell)}(\alpha) := \kappa_0\left(\Sigma_{\text{relu}}^{(\ell-1)}(\alpha)\right). \quad (3)$$

3. Let $K_{\text{relu}}^{(0)}(\alpha) := \Sigma_{\text{relu}}^{(0)}(\alpha) = \alpha$ and for $\ell = 1, 2, \ldots L$, define $K_{\text{relu}}^{(\ell)}(\alpha)$ recursively as follows,

$$K_{\text{relu}}^{(\ell)}(\alpha) := K_{\text{relu}}^{(\ell-1)}(\alpha) \cdot \dot{\Sigma}_{\text{relu}}^{(\ell)}(\alpha) + \Sigma_{\text{relu}}^{(\ell)}(\alpha). \quad (4)$$

The connection between ReLU-NTK function $K_{\text{relu}}^{(L)}$ and the NTK kernel $\Theta_{\text{ntk}}^{(L)}$ is formalized bellow,

$$\Theta_{\text{ntk}}^{(L)}(y, z) \equiv \|y\|_2 \|z\|_2 \cdot K_{\text{relu}}^{(L)}\left(\frac{\langle y, z \rangle}{\|y\|_2 \|z\|_2}\right), \quad \text{for any } y, z \in \mathbb{R}^d. \quad (5)$$

This shows that the NTK is a *normalized dot-product kernel* which can be fully characterized by $K_{\text{relu}}^{(L)} : [-1, 1] \to \mathbb{R}$, plotted in Fig. 1. As shown in Fig. 1, this function is smooth and can be tightly approximated with a low-degree polynomial. It is evident that for larger values of $L$, $K_{\text{relu}}^{(L)}(\cdot)$ converges to a *knee shape*, i.e., it has a nearly constant value of roughly $0.3(L + 1)$ on the interval $[-1, 1 - \mathcal{O}(L^{-1})]$, and on the interval $[1 - \mathcal{O}(L^{-1}), 1]$ its value sharply increases to $L + 1$ at $\alpha = 1$.

---

**Algorithm 1** NTKSKETCH for fully-connected ReLU networks

---
1: **input**: vector $x \in \mathbb{R}^d$, network depth $L$, error and failure parameters $\varepsilon, \delta > 0$
2: Choose integers $s = \widetilde{\mathcal{O}}\left(\frac{L^2}{\varepsilon^2}\right)$, $n_1 = \widetilde{\mathcal{O}}\left(\frac{L^4}{\varepsilon^4}\right)$, $r = \widetilde{\mathcal{O}}\left(\frac{L^6}{\varepsilon^4}\right)$, $m = \widetilde{\mathcal{O}}\left(\frac{L^8}{\varepsilon^{\frac{16}{3}}}\right)$, and $s^* = \mathcal{O}\left(\frac{1}{\varepsilon^2}\log\frac{1}{\delta}\right)$ appropriately[†]
3: For $p = \left\lceil 2L^2/\varepsilon^{\frac{4}{3}}\right\rceil$ and $p' = \left\lceil 9L^2/\varepsilon^2\right\rceil$, polynomials $P_{\text{relu}}^{(p)}(\cdot)$ and $\dot{P}_{\text{relu}}^{(p')}(\cdot)$ are defined as,

$$
\begin{aligned}
P_{\text{relu}}^{(p)}(\alpha) &\equiv \sum_{j=0}^{2p+2} c_j \cdot \alpha^j := \frac{1}{\pi} + \frac{\alpha}{2} + \frac{1}{\pi}\sum_{i=0}^{p} \frac{(2i)! \cdot \alpha^{2i+2}}{2^{2i}(i!)^2(2i+1)(2i+2)}, \\
\dot{P}_{\text{relu}}^{(p')}(\alpha) &\equiv \sum_{j=0}^{2p'+1} b_j \cdot \alpha^j := \frac{1}{2} + \frac{1}{\pi}\sum_{i=0}^{p'} \frac{(2i)!}{2^{2i}(i!)^2(2i+1)} \cdot \alpha^{2i+1}.
\end{aligned}
\tag{6}
$$

4: $\phi^{(0)}(x) \leftarrow \|x\|_2^{-1} \cdot \boldsymbol{Q}^1 \cdot x$, where $\boldsymbol{Q}^1 \in \mathbb{R}^{r \times d}$ is a degree-1 POLYSKETCH as per Lemma 1
5: $\psi^{(0)}(x) \leftarrow \boldsymbol{V} \cdot \phi^{(0)}(x)$, where $\boldsymbol{V} \in \mathbb{R}^{s \times r}$ is an instance of SRHT [2]
6: **for** $\ell = 1$ to $L$ **do**
7:     Let $\boldsymbol{Q}^{2p+2} \in \mathbb{R}^{m \times r^{2p+2}}$ be a degree-$2p + 2$ POLYSKETCH. Also, let $\boldsymbol{T} \in \mathbb{R}^{r \times (2p+3) \cdot m}$ be an instance of SRHT. For every $l = 0, 1, \ldots, 2p + 2$, compute:

$$
Z_l^{(\ell)}(x) \leftarrow \boldsymbol{Q}^{2p+2}\left(\left[\phi^{(\ell-1)}(x)\right]^{\otimes l} \otimes e_1^{\otimes 2p+2-l}\right), \quad \phi^{(\ell)}(x) \leftarrow \boldsymbol{T} \cdot \bigoplus_{l=0}^{2p+2} \sqrt{c_l} Z_l^{(\ell)}(x) \tag{7}
$$

8:     Let $\boldsymbol{Q}^{2p'+1} \in \mathbb{R}^{n_1 \times r^{2p'+1}}$ be a degree-$2p' + 1$ POLYSKETCH. Also, let $\boldsymbol{W} \in \mathbb{R}^{s \times (2p'+2) \cdot n_1}$ be an instance of SRHT. For every $l = 0, 1, \ldots, 2p' + 1$, compute:

$$
Y_l^{(\ell)}(x) \leftarrow \boldsymbol{Q}^{2p'+1}\left(\left[\phi^{(\ell-1)}(x)\right]^{\otimes l} \otimes e_1^{\otimes 2p'+1-l}\right), \quad \dot{\phi}^{(\ell)}(x) \leftarrow \boldsymbol{W} \cdot \bigoplus_{l=0}^{2p'+1} \sqrt{b_l} Y_l^{(\ell)}(x) \tag{8}
$$

9:     Let $\boldsymbol{Q}^2 \in \mathbb{R}^{s \times s^2}$ be a degree-2 POLYSKETCH. Also, let $\boldsymbol{R} \in \mathbb{R}^{s \times (s+r)}$ be an SRHT. Compute:

$$
\psi^{(\ell)}(x) \leftarrow \boldsymbol{R} \cdot \left(\boldsymbol{Q}^2\left(\psi^{(\ell-1)}(x) \otimes \dot{\phi}^{(\ell)}(x)\right) \oplus \phi^{(\ell)}(x)\right). \tag{9}
$$

10: Let $\boldsymbol{G} \in \mathbb{R}^{s^* \times s}$ be a matrix of i.i.d. entries with distribution $\mathcal{N}(0, \frac{1}{s^*})$. Compute:

$$
\Psi_{\text{ntk}}^{(L)}(x) \leftarrow \|x\|_2 \cdot \boldsymbol{G} \cdot \psi^{(L)}(x). \tag{10}
$$

11: **return** $\Psi_{\text{ntk}}^{(L)}(x)$

---

## 3 Sketching and Random Features for NTK

The main results of this section are efficient oblivious sketching as well as random features for the fully-connected NTK. As shown in Definition 1 and Eq. (5), the NTK $\Theta_{\text{ntk}}^{(L)}$, is constructed by recursive composition of arc-cosine kernels $\kappa_1(\cdot)$ and $\kappa_0(\cdot)$. So, to design efficient sketches for the NTK we crucially need efficient methods for approximating these functions. Generally, there are two main approaches to approximating these functions; one is random features sampling and the other is truncated Taylor series expansion coupled with fast sketching. We design algorithms by exploiting both of these techniques.

### 3.1 NTK Sketch

Our main tool is approximating the arc-cosine kernels with low-degree polynomials, and then applying POLYSKETCH to the resulting polynomial kernels. The features for multi-layer NTK are the recursive tensor product of arc-cosine sketches at consecutive layers, which in turn can be sketched efficiently using POLYSKETCH. We present our oblivious sketch in Algorithm 1.

Now we present our main theorem on NTKSKETCH algorithm as follows.

**Theorem 1.** *For every integers $d \geq 1$ and $L \geq 2$, and any $\varepsilon, \delta > 0$, let $\Theta_{\mathrm{ntk}}^{(L)} : \mathbb{R}^d \times \mathbb{R}^d \to \mathbb{R}$ be the L-layer NTK with ReLU activation as per Definition 1 and Eq. (5). Then there exists a randomized map $\Psi_{\mathrm{ntk}}^{(L)} : \mathbb{R}^d \to \mathbb{R}^{s^*}$ for some $s^* = \mathcal{O}\left(\frac{1}{\varepsilon^2} \log \frac{1}{\delta}\right)$ such that the following invariants hold,*

*1. For any vectors $y, z \in \mathbb{R}^d$: $\Pr\left[\left|\left\langle \Psi_{\mathrm{ntk}}^{(L)}(y), \Psi_{\mathrm{ntk}}^{(L)}(z) \right\rangle - \Theta_{\mathrm{ntk}}^{(L)}(y,z)\right| \leq \varepsilon \cdot \Theta_{\mathrm{ntk}}^{(L)}(y,z)\right] \geq 1 - \delta.$*

*2. For every vecor $x \in \mathbb{R}^d$, the time to compute $\Psi_{\mathrm{ntk}}^{(L)}(x)$ is $\mathcal{O}\left(\frac{L^{11}}{\varepsilon^{6.7}} \log^3 \frac{L}{\varepsilon\delta} + \frac{L^3}{\varepsilon^2} \log \frac{L}{\varepsilon\delta} \cdot \mathrm{nnz}(x)\right).$*

For a proof, see Appendix C. One can observe that the runtime of our NTKSKETCH is faster than the gradient features of an ultra-wide random DNN, studied by Arora et al. [5], by a factor of $L^3/\varepsilon^2$.

## 3.2 NTK Random Features

The main difference between our random features construction and NTKSKETCH is the use of random features for approximating arc-cosine kernels $\kappa_0$ and $\kappa_1$ in Eq. (2). For any $x \in \mathbb{R}^d$, we denote

$$\Phi_0(x) := \sqrt{\frac{2}{m_0}} \, \mathrm{Step}\left([w_1, \ldots, w_{m_0}]^\top x\right), \quad \Phi_1(x) := \sqrt{\frac{2}{m_1}} \, \mathrm{ReLU}\left([w'_1, \ldots, w'_{m_1}]^\top x\right), \quad (11)$$

where $w_1, \ldots, w_{m_0}, w'_1, \ldots, w'_{m_1} \in \mathbb{R}^d$ are i.i.d. samples from $\mathcal{N}(0, \boldsymbol{I}_d)$. Cho and Saul [12] showed that $\mathbb{E}[\langle \Phi_0(y), \Phi_0(z) \rangle] = \kappa_0\left(\frac{\langle y,z \rangle}{\|y\|_2 \|z\|_2}\right)$ and $\mathbb{E}[\langle \Phi_1(y), \Phi_1(z) \rangle] = \|y\|_2 \|z\|_2 \cdot \kappa_1\left(\frac{\langle y,z \rangle}{\|y\|_2 \|z\|_2}\right)$. The feature map for multi-layer NTK can be obtained by recursive tensoring of random feature maps for arc-cosine kernels at each layer of the network. However, one major drawback of such explicit tensoring is that the number of features, and thus the runtime, will be exponential in depth $L$. In order to make the feature map more compact, we utilize a degree-2 POLYSKETCH $\boldsymbol{Q}^2$ to reduce the dimension of the tensor products at each layer and get rid of exponential dependence on $L$. We present the performance guarantee of our random features, defined in Algorithm 2, in Theorem 2.

**Theorem 2.** *Given $y, z \in \mathbb{R}^d$ and $L \geq 2$, let $\Theta_{\mathrm{ntk}}^{(L)}$ the L-layer fully-connected ReLU NTK. For $\varepsilon, \delta > 0$, there exist $m_0 = \mathcal{O}\left(\frac{L^2}{\varepsilon^2} \log \frac{L}{\delta}\right), m_1 = \mathcal{O}\left(\frac{L^6}{\varepsilon^4} \log \frac{L}{\delta}\right), m_s = \mathcal{O}\left(\frac{L^2}{\varepsilon^2} \log^3 \frac{L}{\varepsilon\delta}\right)$, such that,*

$$\Pr\left[\left|\left\langle \Psi_{\mathrm{rf}}^{(L)}(y), \Psi_{\mathrm{rf}}^{(L)}(z) \right\rangle - \Theta_{\mathrm{ntk}}^{(L)}(y,z)\right| \leq \varepsilon \cdot \Theta_{\mathrm{ntk}}^{(L)}(y,z)\right] \geq 1 - \delta, \quad (12)$$

*where $\Psi_{\mathrm{rf}}^{(L)}(y), \Psi_{\mathrm{rf}}^{(L)}(z) \in \mathbb{R}^{m_1 + m_s}$ are the outputs of Algorithm 2, using the same randomness.*

The proof of Theorem 2 is provided in Appendix D. Arora et al. [5] proved that the gradient of randomly initialized ReLU network with finite width can approximate the NTK, but their feature dimension should be $\Omega\left(\frac{L^{13}}{\varepsilon^8} \log^2 \frac{L}{\delta} + \frac{L^6}{\varepsilon^4} \cdot \log \frac{L}{\delta} \cdot d\right)$ which is larger than ours by a factor of $\frac{L^7}{\varepsilon^4} \log \frac{L}{\delta}$. In Section 5, we also empirically show that Algorithm 2 requires far fewer features than random gradients.

## 3.3 Spectral Approximation for NTK via Leverage Scores Sampling

Although the above NTK approximations can estimate the kernel function itself, it is still questionable how it affects the performance of downstream tasks. Several works on kernel approximation adopt spectral approximation bound with regularization $\lambda > 0$ and approximation factor $\varepsilon > 0$, that is,

$$(1 - \varepsilon)(\boldsymbol{K}_{\mathrm{ntk}}^{(L)} + \lambda \boldsymbol{I}) \preceq (\boldsymbol{\Psi}^{(L)})^\top \boldsymbol{\Psi}^{(L)} + \lambda \boldsymbol{I} \preceq (1 + \varepsilon)(\boldsymbol{K}_{\mathrm{ntk}}^{(L)} + \lambda \boldsymbol{I}), \quad (13)$$

where $\boldsymbol{\Psi}^{(L)} := \left[\Psi^{(L)}(x_1), \ldots, \Psi^{(L)}(x_n)\right]$ and $[\boldsymbol{K}_{\mathrm{ntk}}^{(L)}]_{i,j} = \Theta_{\mathrm{ntk}}^{(L)}(x_i, x_j)$. The spectral bound can provide rigorous guarantees for downstream applications including kernel ridge regression [8], clustering and PCA [32]. We first provide spectral bounds for arc-cosine kernels, then we present our spectral approximation bound for two-layer ReLU networks, which is the first in the literature.

---

[†] $\widetilde{\mathcal{O}}(\cdot)$ suppresses $\mathrm{poly}(\log \frac{L}{\varepsilon\delta})$ factors.

---

**Algorithm 2** Random Features for ReLU NTK via POLYSKETCH

---

1: **input**: vector $x \in \mathbb{R}^d$, network depth $L$, feature dimensions $m_0$, $m_1$, and $m_s$
2: $\psi_{\text{rf}}^{(0)}(x) \leftarrow x/\|x\|_2$, $\phi_{\text{rf}}^{(0)}(x) \leftarrow x/\|x\|_2$
3: **for** $\ell = 1$ to $L$ **do**
4: $\quad \dot{\phi}_{\text{rf}}^{(\ell)}(x) \leftarrow \Phi_0\left(\phi_{\text{rf}}^{(\ell-1)}(x)\right)$, where $\Phi_0$ is defines as per Eq. (11) with $m_0$ features
5: $\quad \phi_{\text{rf}}^{(\ell)}(x) \leftarrow \Phi_1\left(\phi_{\text{rf}}^{(\ell-1)}(x)\right)$, where $\Phi_1$ is defines as per Eq. (11) with $m_1$ features
6: $\quad$ Draw a degree-2 POLYSKETCH $\boldsymbol{Q}^2$ that maps to $\mathbb{R}^{m_s}$ and compute:

$$\psi_{\text{rf}}^{(\ell)}(x) \leftarrow \phi_{\text{rf}}^{(\ell)}(x) \oplus \boldsymbol{Q}^2 \cdot \left(\dot{\phi}_{\text{rf}}^{(\ell)}(x) \otimes \psi_{\text{rf}}^{(\ell-1)}(x)\right)$$

7: **return** $\Psi_{\text{rf}}^{(L)}(x) \leftarrow \|x\|_2 \cdot \psi_{\text{rf}}^{(L)}(x)$

---

To guarantee that the arc-cosine random features in Eq. (11) provide spectral approximation, we will use the leverage score sampling framework of [8, 28]. We reduce the variance of random features by performing importance sampling. The challenge is to find a proper modified distribution that certainly reduces the variance. It turns out that the original $0^{th}$ order arc-cosine random features has a small enough variance. More precisely, let $\boldsymbol{K}_0$ be the $0^{th}$ order arc-cosine kernel matrix, i.e., $[\boldsymbol{K}_0]_{i,j} = \kappa_0\left(\frac{\langle x_i, x_j \rangle}{\|x_i\|_2 \|x_j\|_2}\right)$, and $\boldsymbol{\Phi}_0 := [\Phi_0(x_1), \ldots, \Phi_0(x_n)]$, where $\Phi_0(x)$ is defined in Eq. (11). If the number of features $m_0 \geq \frac{8}{3}\frac{n}{\lambda \varepsilon^2} \log\left(\frac{16s_\lambda}{\delta}\right)$, then

$$\Pr\left[(1-\varepsilon)(\boldsymbol{K}_0 + \lambda \boldsymbol{I}) \preceq \boldsymbol{\Phi}_0^\top \boldsymbol{\Phi}_0 + \lambda \boldsymbol{I} \preceq (1+\varepsilon)(\boldsymbol{K}_0 + \lambda \boldsymbol{I})\right] \geq 1 - \delta. \tag{14}$$

Next, we consider spectral approximation of the $1^{st}$ order arc-cosine kernel. Unlike the previous case, modifications of the sampling distribution are required. Specifically, for any $x \in \mathbb{R}^d$, let

$$\widetilde{\Phi}_1(x) = \sqrt{\frac{2d}{m_1}} \text{ReLU}\left(\left[\frac{w_1}{\|w_1\|_2}, \ldots, \frac{w_{m_1}}{\|w_{m_1}\|_2}\right]^\top x\right), \tag{15}$$

where $w_1, \ldots, w_{m_1} \in \mathbb{R}^d$ are i.i.d. samples from $p(w) := \frac{1}{(2\pi)^{d/2}d} \|w\|_2^2 \exp\left(-\frac{1}{2}\|w\|_2^2\right)$. For this modified features, let $\boldsymbol{X} \in \mathbb{R}^{d \times n}$ be the dataset, $\boldsymbol{K}_1$ be the $1^{st}$ order arc-cosine kernel matrix, i.e., $[\boldsymbol{K}_1]_{i,j} = \|x_i\|_2 \|x_j\|_2 \cdot \kappa_1\left(\frac{\langle x_i, x_j \rangle}{\|x_i\|_2 \|x_j\|_2}\right)$, and $\boldsymbol{\Phi}_1 := \left[\widetilde{\Phi}_1(x_1), \ldots, \widetilde{\Phi}_1(x_n)\right]$. If the number of features $m_1 \geq \frac{8}{3}\frac{d}{\varepsilon^2} \cdot \min\left\{\text{rank}(\boldsymbol{X})^2, \frac{\|\boldsymbol{X}\|_2^2}{\lambda}\right\} \log\left(\frac{16s_\lambda}{\delta}\right)$, then

$$\Pr\left[(1-\varepsilon)(\boldsymbol{K}_1 + \lambda \boldsymbol{I}) \preceq \boldsymbol{\Phi}_1^\top \boldsymbol{\Phi}_1 + \lambda \boldsymbol{I} \preceq (1+\varepsilon)(\boldsymbol{K}_1 + \lambda \boldsymbol{I})\right] \geq 1 - \delta. \tag{16}$$

The details are provided in Appendix E.1 and Appendix E.2. We are now ready to state our spectral approximation bound for our modified random features.

**Theorem 3.** *Given a dataset $\boldsymbol{X} \in \mathbb{R}^{d \times n}$ with $\|\boldsymbol{X}_{(:,i)}\|_2 \leq 1$ for every $i \in [n]$, let $\boldsymbol{K}_{\text{ntk}}, \boldsymbol{K}_0, \boldsymbol{K}_1$ be kernel matrices for two-layer ReLU NTK and arc-cosine kernels of $0^{th}$ and $1^{st}$ order, respectively. For any $\lambda > 0$, suppose $s_\lambda$ is the statistical dimension of $\boldsymbol{K}_{\text{ntk}}$. Modify Algorithm 2 by replacing $\Phi_1(\cdot)$ in line 5 with $\widetilde{\Phi}_1(\cdot)$ defined in Eq. (15). For any $\varepsilon, \delta > 0$, let $\boldsymbol{\Psi}_{\text{rf}}^{(L)} \in \mathbb{R}^{(m_1 + m_s) \times n}$ be the output matrix of this algorithm with $L = 1$. There exist $m_0 = \mathcal{O}\left(\frac{n}{\varepsilon^2 \lambda} \log \frac{s_\lambda}{\delta}\right), m_1 = \mathcal{O}\left(\frac{d}{\varepsilon^2} \cdot \min\left\{\text{rank}(\boldsymbol{X})^2, \frac{\|\boldsymbol{X}\|_2^2}{\lambda}\right\} \log \frac{s_\lambda}{\delta}\right), m_s = \mathcal{O}\left(\frac{1}{\varepsilon^2} \cdot \frac{n}{1+\lambda} \log^3 \frac{n}{\varepsilon \delta}\right)$ such that,*

$$\Pr\left[(1-\varepsilon)\left(\boldsymbol{K}_{\text{ntk}} + \lambda \boldsymbol{I}\right) \preceq \left(\boldsymbol{\Psi}_{\text{rf}}^{(L)}\right)^\top \boldsymbol{\Psi}_{\text{rf}}^{(L)} + \lambda \boldsymbol{I} \preceq (1+\varepsilon)\left(\boldsymbol{K}_{\text{ntk}} + \lambda \boldsymbol{I}\right)\right] \geq 1 - \delta. \tag{17}$$

For a proof see Appendix E.3. To generalize the current proof technique to deeper networks, one needs a monotone property of arc-cosine kernels, i.e., $\kappa_1(\boldsymbol{X}) \preceq \kappa_1(\boldsymbol{Y})$ for $\boldsymbol{X} \preceq \boldsymbol{Y}$. However, this property does not hold in general and we leave the extension to deeper networks to future work.

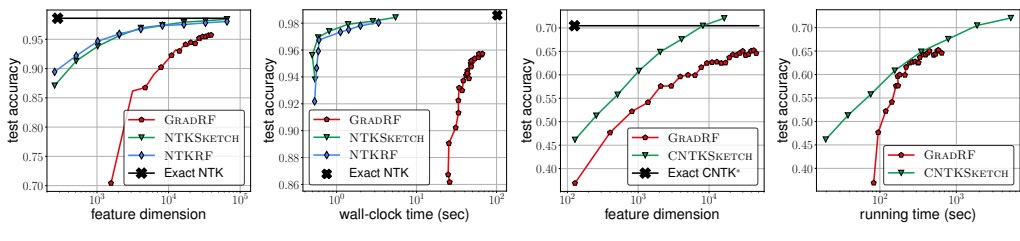

(a) MNIST with NTK        (b) CIFAR-10 with CNTK

Figure 2: Test accuracy of: (a) approximate NTK methods (GRADRF, NTKSKETCH and NTKRF) on MNIST and (b) approximate CNTK methods (GRADRF and CNTKSKETCH) on CIFAR-10.

## 4 Sketching Convolutional Neural Tangent Kernel

In this section, we design and analyze an efficient sketching method for the Convolutional Neural Tangent Kernel (CNTK). We focus mainly on CNTK with Global Average Pooling (GAP), which exhibits superior empirical performance compared to vanilla CNTK with no pooling [5], however, our techniques can be applied to the vanilla version, as well. Using the DP of Arora et al. [5], the number of operations needed for exact computation of the depth-$L$ CNTK value $\Theta_{\text{cntk}}^{(L)}(y, z)$ for images $y, z \in \mathbb{R}^{d \times d}$ is $\Omega\left(d^4 \cdot L\right)$, which is extremely slow particularly due to its quadratic dependence on the number of pixels of input images $d^2$. Fortunately, we are able to show that the CNTK for the important case of ReLU activation is a highly structured object that can be fully characterized in terms of tensoring and composition of arc-cosine kernels, and exploiting this special structure is key to designing efficient sketching methods for the CNTK. Unlike the fully-connected NTK, CNTK is not a simple dot-product kernel function like Eq. (5). The key reason being that CNTK works by partitioning its input images into patches and locally transforming the patches at each layer, as opposed to the NTK which operates on the entire input vectors. We present our derivation of the ReLU CNTK function and its main properties in Appendix F.

Similar to NTKSKETCH our method relies on approximating the arc-cosine kernels with low-degree polynomials via Taylor expansion, and then applying POLYSKETCH to the resulting polynomial kernels. Our sketch computes the features for each pixel of the input image, by tensor product of arc-cosine sketches at consecutive layers, which in turn can be sketched efficiently using POLYSKETCH. Additionally, the features of pixels that lie in the same patch get *locally combined* at each layer via direct sum operation. This precisely corresponds to the convolution operation in neural networks. We present our CNTKSKETCH algorithm in Appendix G and give its performance guarantee in the following theorem.

**Theorem 4.** *For every positive integers $d_1, d_2, c$ and $L \geq 2$, and every $\varepsilon, \delta > 0$, if we let $\Theta_{\text{cntk}}^{(L)} : \mathbb{R}^{d_1 \times d_2 \times c} \times \mathbb{R}^{d_1 \times d_2 \times c} \to \mathbb{R}$ be the $L$-layer CNTK with ReLU activation and GAP given in [5], then there exist a randomized map $\Psi_{\text{cntk}}^{(L)} : \mathbb{R}^{d_1 \times d_2 \times c} \to \mathbb{R}^{s^*}$ for some $s^* = \mathcal{O}\left(\frac{1}{\varepsilon^2} \log \frac{1}{\delta}\right)$ such that:*

*1. For any images $y, z \in \mathbb{R}^{d_1 \times d_2 \times c}$:*

$$\Pr\left[\left|\left\langle \Psi_{\text{cntk}}^{(L)}(y), \Psi_{\text{cntk}}^{(L)}(z) \right\rangle - \Theta_{\text{cntk}}^{(L)}(y, z)\right| \leq \varepsilon \cdot \Theta_{\text{cntk}}^{(L)}(y, z)\right] \geq 1 - \delta.$$

*2. For every image $x \in \mathbb{R}^{d_1 \times d_2 \times c}$, time to compute $\Psi_{\text{cntk}}^{(L)}(x)$ is $\mathcal{O}\left(\frac{L^{11}}{\varepsilon^{6.7}} \cdot (d_1 d_2) \cdot \log^3 \frac{d_1 d_2 L}{\varepsilon \delta}\right)$.*

The proof is in Appendix G. Runtime of our CNTKSKETCH is only linear in the number of image pixels $d_1 d_2$, which is in stark contrast to quadratic scaling of the exact CNTK computation [5].

## 5 Experiments

In this section, we empirically show that running least squares regression on the features generated by our methods is extremely fast and effective for learning with NTK and CNTK kernel machines. We run experiments on a system with an Intel E5-2630 CPU with 256 GB RAM and a single GeForce RTX 2080 GPUs with 12 GB RAM. Codes are available at https://github.com/insuhan/ntk-sketch-rf.

Table 1: Test accuracy and runtime to solve CNTK regression and its approximations on CIFAR-10. (*) means that the result is copied from Arora et al. [5].

| | CNTKSKETCH | (ours) | | GRADRF | | | Exact CNTK | CNN |
|---|---|---|---|---|---|---|---|---|
| Feature dimension | 4,096 | 8,192 | 16,384 | 9,328 | 17,040 | 42,816 | | |
| Test accuracy (%) | 67.58 | 70.46 | 72.06 | 62.49 | 62.57 | 65.21 | 70.47* | 63.81* |
| Time (s) | 780 | 1,870 | 5,160 | 300 | 360 | 580 | > 1,000,000 | |

Table 2: MSE and runtime on large-scale UCI datasets. We measure the entire time to solve kernel ridge regression. $(-)$ means Out-of-Memory error.

| | MillionSongs | | WorkLoads | | CT | | Protein | |
|---|---|---|---|---|---|---|---|---|
| # of data points ($n$) | 467,315 | | 179,585 | | 53,500 | | 39,617 | |
| | MSE | Time (s) | MSE | Time (s) | MSE | Time (s) | MSE | Time (s) |
| RBF Kernel | − | − | − | − | 35.37 | 59.23 | 18.96 | 46.45 |
| RFF | 109.50 | 231 | $4.034 \times 10^4$ | 53.0 | 48.20 | 15.2 | 19.72 | 12.1 |
| NTK | − | − | − | − | 30.52 | 72.10 | 20.24 | 76.93 |
| NTKRF (ours) | 94.27 | 95 | $3.554 \times 10^4$ | 35.7 | 46.91 | 2.12 | 20.51 | 4.3 |
| NTKSKETCH (ours) | 92.83 | 36 | $3.538 \times 10^4$ | 27.5 | 46.52 | 18.8 | 21.19 | 14.91 |

## 5.1 NTK Classification on MNIST

We first benchmark our proposed NTK approximation algorithms on MNIST [25] dataset and compare against gradient-based NTK random features [5] (GRADRF) as a baseline method. To apply our methods and GRADRF into classification task, we encode class labels into one-hot vectors with zero-mean and solve the ridge regression problem. We search the ridge parameter with a random subset of training set and choose the one that achieves the best validation accuracy. We use the ReLU network with depth $L = 1$. In Fig. 2a, we observe that our random features (NTKRF ) achieves the best test accuracy. The NTKSKETCH narrowly follows the performance of NTKRF and the Grad-RF is the worst method which confirms the observations of Arora et al. [5], i.e., gradient of a finite width network degrades practical performances.

**Remark 1** (Optimizing NTKSKETCH for Deeper Nets). As shown in Eq. (5), the NTK is a normalized dot-product kernel characterized by the function $K_{\texttt{relu}}^{(L)}(\alpha)$. This function can be easily computed using $\mathcal{O}(L)$ operations at any desired $\alpha \in [-1, 1]$, therefore, we can efficiently fit a polynomial to this function using numerical methods (for instance, it is shown in Fig. 1 that a degree-8 polynomial can tightly approximate the depth-3 ReLU-NTK function $K_{\texttt{relu}}^{(3)}$). Then, we can efficiently sketch the resulting polynomial kernel using POLYSKETCH , as was previously done for Gaussian and general dot-product kernels [1, 40]. Therefore, we can accelerate our NTKSKETCH for deeper networks ($L > 2$), using this heuristic.

## 5.2 CNTK Classification on CIFAR-10

Next we test our CNTKSKETCH on CIFAR-10 dataset [24]. We choose a convolutional network of depth $L = 3$ and compare CNTKSKETCH and GRADRF for various feature dimensions. We borrow results of both CNTK and CNN from Arora et al. [5]. The results are provided in Fig. 2b and Table 1. Somewhat surprisingly, CNTKSKETCH even performs better than the exact CNTK regression by achieving $72.06\%$ when feature dimension is set to 16,384. The likely explanation is that CNTKSKETCH takes advantages of implicit regularization effects of approximate feature map and powerful expressiveness of the CNTK. Moreover, computing the CNTK matrix takes over 250 hours (12 days) under our setting which is at least $150\times$ slower than our CNTKSKETCH.

## 5.3 Regression on Large-scale UCI Datasets

We also demonstrate the computational efficiency of our NTKSKETCH and NTKRF using 4 large-scale UCI regression datasets [17] by comparing against exact NTK, RBF as well as Random Fourier Features (RFF). For our methods and RFF, we fix the output dimension to $m = 8,192$ for all datasets. In Table 2, we report the runtime to compute feature map or kernel matrix and evaluate the averaged mean squared errors (MSE) on the test set via 4-fold cross validation. The exact kernel methods face

Out-of-Memory error on larger datasets. The proposed NTK approximations are significantly faster than the exact NTK, e.g., NTKRF shows up to $30\times$ speedup under $\mathrm{CT}$ dataset. We also verify that, except for $\mathrm{Protein}$ dataset, our methods outperform RFF.

## 6    Discussion and Conclusion

In this work, we propose efficient low-rank feature maps for the NTK and CNTK kernel matrices based on both sketching and random features. Computing NTK have been raised severe computational problems when they apply to practical applications. Our methods runs remarkably faster than the NTK with performance improvement.

**Potential negative societal impact.** This is a technical work proposing provable algorithms which stand alone independently of data, e.g., do not learn any private information of input data. We think there is no particular potential negative societal impact due to our work.

**Limitations.** This paper only considers fully-connected and convolutional neural networks, and our ideas are not directly applicable to scale up NTK of other deep networks, e.g., transformers [21].

## Acknowledgments and Disclosure of Funding

Amir Zandieh was partially supported by the Swiss NSF grant No. P2ELP2_195140. Haim Avron and Neta Shoham were partially supported by BSF grant 2017698 and ISF grant 1272/17. Jinwoo Shin was partially supported by the Engineering Research Center Program through the National Research Foundation of Korea (NRF) funded by the Korean Government MSIT (NRF_2018R1A5A1059921) and Institute of Information & communications Technology Planning & Evaluation (IITP) grant funded by the Korea government (MSIT) (No.2019_0_00075, Artificial Intelligence Graduate School Program (KAIST)).

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
