# OpenReview forum: "Scaling Neural Tangent Kernels via Sketching and Random Features"
_NeurIPS.cc/2021/Conference — NeurIPS 2021 Poster_

### Official Review · Reviewer_Mq45 · 2021-07-14

**Rating:** 6
**Confidence:** 3

**Summary:**

The submission proposed an low-rank feature maps method to accelerate learning with the Neural Tangent Kernel (NTK). They utlized the sketching and random features to realize the approximation. A theoretical analysis from the spectral pespertive has been applied. Finally, experiments on large-scale regression and classification tasks show the time-efficent of the proposed method.

**Limitations And Societal Impact:**

 The authors adequately addressed the limitations and potential negative societal impact of their work

**Main Review:**

Originality: Approximating the NTK for the large-scale task is somewhat novel to me. The related works are adequately cited and it is clear how this work differs from the previous contribution according to the presentation by authors.

Quality: The submission is technically sound and all claims including theoretical analysis and experimental results are well supported. However, I have some concerns regarding the weaknesses of this work:

(1) This work only considers arc-cosine kernel (corresponding to ReLU activation), can you extend to more kernels (activations)?
(2) This work only considers NTK for fully connected networks and CNTK, which you have mentioned in the main text.
(3) The classification and regression tasked used in this work is somewhat fundamental, I would expect more specific but significant application.

Clarity: The submission is clearly written and well organized.

Significance: Since the (C)NTK regression usually performs worse than its counterpart neural network on computer vision tasks [1,2,3]. Thus the significance of accelerating NTK learning for computer vision tasks is somewhat weak to me and relevant to researchers in subareas only.

[1] Sanjeev Arora, Simon S Du, Wei Hu, Zhiyuan Li, Ruslan Salakhutdinov, and Ruosong Wang. On exact computation with an infinitely wide neural net. In NeurIPS. arXiv preprint arXiv:1904.11955.

[2] Ghorbani, Behrooz, et al. "Limitations of lazy training of two-layers neural networks." arXiv preprint arXiv:1906.08899 (2019).

[3] Lee, Jaehoon, et al. "Finite versus infinite neural networks: an empirical study." arXiv preprint arXiv:2007.15801 (2020).


**Time Spent Reviewing:**

3

---

> ### Author Response · Authors · 2021-08-10
> **Authors response**
>
> We thank the reviewer for the detailed and insightful comments. Regarding the comment,
>
> - Extension to other activations:
>   * Although we mainly focus on the ReLU activation because of its popularity, we believe that our methods can be generalized to other activations. Our NTK Random Features for instance can be readily generalized to any differentiable activation function $\sigma(\cdot)$ by replacing the ReLU and Step functions in Eq (11) by $\sigma(.)$ and its derivative $\dot{\sigma}(\cdot)$, respectively.
> Furthermore, our NTKSketch Talyor expands the kernel functions that correspond to the activation function and its derivative (which are arc-cosine kernels of order 1 and 0 in case of ReLU activation) and applies the PolySketch to the corresponding polynomial. Therefore, to generalize it to any activation $\sigma(\cdot)$, it is enough to replace the polynomials $P_{relu}^{(p)}(\cdot)$ and $\dot{P}^{(p)}_{relu}(\cdot)$ in Eq (6) with truncated Taylor expansions of the kernel functions obtained by replacing ReLU and Step in Eq (21) in Appendix A by $\sigma(\cdot)$ and $\dot{\sigma}(\cdot)$.
> For example, in the case of exponential activation, the corresponding kernels that emerge from Eq (21) will be RBF kernel functions. Please see Table 1 in [Daniely et al., 2016] for other activation functions.
>
>     [Daniely., 2016] Daniely, Amit, Roy Frostig, and Yoram Singer. "Toward deeper understanding of neural networks: The power of initialization and a dual view on expressivity." NeurIPS. 2016.
>
>
> - Focus on NTK and CNTK:
>   * We focused on NTK of fully-connected NN and CNTK as they are the most popular deep learning models. It is an exciting open question to extend our method to other deep models such as transformers.
>
> - More specific but significant applications:
>   * Our main contributions are algorithmic design of fast NTK approximation methods with rigorous theoretical guarantees. Our experimental results mainly serve as a demonstration that our theoretical contributions can achieve promising empirical results on real-world datasets. We believe that, as computing the NTK is a fundamental task, our results can be applied to a variety of applications, e.g., neural architecture search [Chen et al., 2021]. We will discuss more practical applications in our final draft and possibly add some new experimental results.
>
>     [Chen et al., 2021] Chen, Wuyang, Xinyu Gong, and Zhangyang Wang. Neural Architecture Search on Imagenet in Four GPU Hours: A Theoretically Inspired Perspective. ICLR. 2021.

---

> > ### Comment · Reviewer_Mq45 · 2021-08-12
> > **Follow up**
> >
> > I would like to thank the authors for the answers to my questions. Their rebuttal clarified some of the issues I addressed in my review.
> >
> > Here are some suggestions:
> >
> > **Extension to other activations.** A relevant reference is neural-tangents [1], where they built a high-level neural network API for specifying complex networks with respect to the NTK. Different kinds of architecture including the choice of activation are supported. Thus, I think this work can be extended to be more practical and beneficial to the researcher in this area.
> >
> > **Extend to other deep models such as transformers.** There is an NTK work for transformer [2], I hope the authors could make a discussion regarding this.
> >
> > Based on the response and comments from other reviewers, even this work only considers vanilla Fully-connected network and CNN, it has a large potential to be more practical, such as transformer, and neural architecture search. Besides, the work itself is technically sound and well written. Thus, I raise my score to 6.
> >
> > [1] https://github.com/google/neural-tangents
> > [2] Hron, Jiri, et al. "Infinite attention: NNGP and NTK for deep attention networks." International Conference on Machine Learning. PMLR, 2020.

---

### Official Review · Reviewer_M4ry · 2021-07-16

**Rating:** 7
**Confidence:** 3

**Summary:**

This paper proposes a near input-sparsity time approximation method to scale up the NTK based on sketching and random feature. Besides, the proposed method can be extended for efficient CNTK computation with linear running time. The theoretical analysis and experimental results show the effectiveness of the proposed method.

**Limitations And Societal Impact:**

Yes

**Main Review:**

Strength:

1. This paper proposed an interesting and useful method to accelerate learning with NTK. It also provides nontrivial theoretical analysis of the proposed method.
2. The proposed method achieves efficient NTK and CNTK computation based on random features. Their experiment results show that the proposed feature mapping can greatly reduce the running time for NTK and CNTK with performance improvement.

Weakness:

- The writing of this paper is not clear enough.  For example, in Algorithm 1,  $P_{relu}^{(p)}(\cdot)$ appears with few explanations within the main content.  Although I can find more information in the supplementary material.
- It seems there is a typo in Theorem 2.  In **Theorem 2**, the probability might be $\leq \delta$.

Questions:

- In Table 2, the running time for NTKRF is much smaller than NTKSKETCH in the latter two datasets while the opposite results can be observed in the first two datasets. Are there some insights into this? Are there any criteria for choosing between these two methods?

- According to the results in (Arora et al.), the result of exact CNTK and CNN is based on setting the depth to 3. If the depth becomes 4, exact CNTK and CNN achieve the accuracy of 75.93 and 80.94, which greatly outperform the proposed method regarding accuracy. What are the results f the proposed method uses the larger feature dimension?

**Time Spent Reviewing:**

6

---

> ### Author Response · Authors · 2021-08-10
> **Authors response**
>
> We thank the reviewer for the detailed and insightful comments. We will fix the typos that were pointed out and also make sure to clearly define and explain all components of our algorithms such as $P^{(p)}_{relu}(\cdot)$. Regarding the comment,
>
> - Running time in Table2:
>   * The NTKSketch has a hyperparameter which is the degree of the polynomial that we use to approximate the arc-cosine kernels. We find the right value for this parameter by hyperparameter search and cross-validation. In the first two datasets, the degree of these polynomials is set to 2 while the latter two use degree 4. This is the main reason NTKSketch runs faster on the first two datasets.
>
>
> - Depth with $L\geq3$:
>   * In our algorithm, the intermediate sketching dimensions and the degree of polynomials we use to approximate the arc-cosine kernels need to grow with the depth $L$. As a result, our runtime scales as $L^{11}$. Consequently, when we run our algorithm for depths $L > 3$ our runtime becomes significantly slower than the case of $L=3$ while its accuracy increases only slightly. Thus, we achieve the best accuracy vs runtime trade-off for $L=3$, so we decided to base our results in Table 1 on $L=3$. We will make sure to include some results for larger depths such as $L=4$ and explain why it runs slower than $L=3$.

---

### Official Review · Reviewer_98MV · 2021-07-24

**Rating:** 7
**Confidence:** 4

**Summary:**

The aim of this work is to design approximation algorithms for Neural Tangent Kernel (NTK) learning to reduce the computational cost. To approximate the ReLU NTK, one approach is to approximate the arc-cosine kernel with low-degree polynomials and further apply the norm-preserving dimensionality reduction on the resulting polynomial kernels. The other approach is based on efficiently reducing the tensor product dimension for random features. To approximate the Convolutional NTK (CNTK), a similar sketching-based approach is proposed with additional techniques to do local combinations of same-patch features. The proposed approaches are measured empirically on MNIST, CIFAR-10, and large-scale UCI datasets.

**Ethical Concerns:**

There is no ethical issue with this paper.

**Limitations And Societal Impact:**

Yes.

**Main Review:**

[pros]
This work is significant given the growing interest in NTK-based models and the need to run NTK on large-scale datasets. The proposed random-feature-based approximation and sketching-based approximation for NTK, as well as the sketching approach for CNTK, all come with theoretical guarantees on approximation and thorough empirical evaluations on real-world datasets.

[cons]
- while both the sketching technique and the random feature technique for NTK have similar approximation guarantees, and also very similar empirical performance in Fig. 2 and Table 2, I would appreciate more qualitative analysis and discussions on when one technique would be more preferable than the others.
- I wonder why using a shallow ReLU network with depth one in experiments presented in Fig. 2 since the theoretical analysis (Thm.1 and 2) are based on assumption L >= 2. It would be more interesting to see the empirical evaluations on cases when there are theoretical guarantees.
- a natural question arises while reading this work: is there a random-feature based approximation for CNTK as a counterpart of what presented in Sec. 3.2? Since it's not mentioned in this work, I wonder if there are some difference between NTK and CNTK that prevents the random-feature based approach to generalize.
- Some suggestions on improving clarity:
	- For the Eq. 12 in Thm. 2, should the \geq be \leq? I believe it's a typo.
	- to be more self-contained: to include the formal definition of NTK and CNTK.
	- in the first sentence of Sec. 2, the notation \Theta and L are used before definition.
	- the second invariant in Thm. 1 use the notation nnz without definition.
	- The \notin notation in the inequality in Thm. 4 is confusing since the term (1 +- epsilon)*\Theta is clearly not a set.

**Time Spent Reviewing:**

5

---

> ### Author Response · Authors · 2021-08-10
> **Authors response**
>
> We thank the reviewer for the detailed comments. Regarding the comment,
>
> - When one technique would be more preferable than the others?
>   * For the running time, NTKSketch is theoretically faster than NTKRF, but practical running time depends on the choice of hyperparameters. For instance, one of the hyperparameters of the NTKSketch is the degree of the polynomials we use to approximate the arc-cosine kernels. We suggest running hyperparameter search for both methods, and if the parameters of NTKSketch turn out to be large, NTKRF would be faster.
>
>
>
> - Why shallow depth?
>   * We like to first clarify something about the statements of Theorems 1 and 2. The only reason we stated these theorems for $L>1$ is that in this case we get a multiplicative $(1+\varepsilon)$ approximation guarantee while for the case of $L=1$ we get an additive $\varepsilon$ error. In fact, this has only to do with the fact that the NTK kernel function is bounded away from zero for any $L>1$ while for $L=1$ the NTK can cross zero or even take negative values. There is no fundamental reason for stating our results for $L>1$ and we make sure to modify the theorem statements to include the additive $\varepsilon$ error bound for $L=1$. Regarding the experiments, we chose the kernel depth value by running cross validation on the training set and it just happened that $L=1$ turned out to work best for all datasets.
>
>
>
> - Random-feature based approximation for CNTK:
>   * It is definitely possible to extend the proposed random feature method to CNTK. We decided to only present the CNTK Sketch because it has a faster runtime than the random feature method for CNTK. We make sure to mention the possibility of the CNTK random features in our final draft.
>
> We will also update all suggestions for improving the readability in our final draft.

---

> > ### Comment · Reviewer_98MV · 2021-08-30
> > **Thanks**
> >
> > I'd like to thank the authors for their response. It answers some concerns that I raised in the detailed comments. My score for this work remains unchanged and I'm inclined to see its acceptance.

---

### Official Review · Reviewer_Rtas · 2021-07-26

**Rating:** 6
**Confidence:** 1

**Summary:**

In this paper, a sketching algorithm for the Neural Tangent Kernel (NTK) is proposed to accelerate its computation. Besides, a proof of the spectral approximation guarantee for the NTK matrix is provided, which combines the random features of the arc-cosine kernels with a sketching algorithm. Empirical studies on the MNIST and CIFAR-10 image classification task show that the linear regressor trained the sketch CNTK features matches the accuracy of the exact CNTK while being 150x faster.

**Limitations And Societal Impact:**

They indeed discussed the limitation of the methods. For example, they cannot be applied to transformers right now.

There would not be negative societal impact of the work.

**Main Review:**

The recent progress on the Neural Tangent Kernel (NTK) builds a bridge between the neural network and kernel learning, making it worthwhile to investigate efforts to study the property of NTK. The NTK and the traditional kernel methods suffers the same bottleneck, i.e., the computational complexity, which naturally inspires this paper to adopt the sketching ideas to accelerate the kernel computation.

Although I did not go through all the equations of the paper, the success in the sketching idea in the traditional kernel learning makes me believe it can transfer to the NTK setting, and the experimental section confirms this point. In fact, the speedup of the proposed methods shown in Table 1 and 2 is quite cheerful, making the training of CNTK more practical than before.

One question: is it possible to apply the idea in the following paper to make it even faster?

[1] Fastfood — Approximating Kernel Expansions in Loglinear Time. Quoc Le, Tamas Sarlos and Alex Smola. ICML 2013.

**Time Spent Reviewing:**

3

---

> ### Author Response · Authors · 2021-08-10
> **Authors response**
>
> We thank the reviewer for the detailed and insightful comments. Regarding the applicability of ideas from the Fastfood algorithm [Quoc et al., 2013], we indeed use a generalized version of these ideas in our methods in the following ways. The underlying idea of Fastfood is based on sampling the random features of a kernel function and then reducing the dimensionality of features using the Subsampled Randomized Hadamard Transform (SRHT). Our method, on the other hand, generates features for the arc-cosine kernels (by either sampling random features or sketching its polynomial expansion) and then combines the resulting features using PolySketch at each layer of the network. The PolySketch is in fact a generalization of SRHT which can be applied to tensor products efficiently. Note that naive application of the Fastfood to the NTK would be very slow because this kernel is a product of various arc-cosine kernels in different layers of the network and thus the features of this kernel, which are the tensor product of the features of arc-cosine kernels, have exponential dimension in the number of layers.
>
> The details of PolySketch and its connections to SRHT are provided in Section B in the supplementary material.
>
> [Quoc et al., 2013] Fastfood — Approximating Kernel Expansions in Loglinear Time.

---

> > ### Comment · Reviewer_Rtas · 2021-08-30
> > **Thanks for the response**
> >
> > I read the authors' response and also other reviews. I think we all agree that this is a good paper and should be accepted. So I won't change my score.

---

### Official Review · Reviewer_hkr8 · 2021-07-27

**Rating:** 7
**Confidence:** 4

**Summary:**

Neural Tangent Kernel (NTK) is a recently developed theoretical tools to analyze the optimization of deep neural network under grandient descent. In particular, the infinitely wide neural network has been shown to be equivalent to kernel regression with NTK. However, a key issue of the NTK in practice is the scalability. This paper aims to tackle this problem by approximating the NTK with sketching and random features. Specifically, the authors propose the following two different approximation algorithms.

The first algorithm proposes to approximate the arc-cosine kernels with low degree plynomials combined with polynomial sketching in order to further reduce the dimensionality of the high-degree tensor products. This algorithm is later applied to CNTK with some modification to cope with the GAP. The authors also provide a detailed theoretical analysis on the approximation error of the NTK.

The second algorithm utilizes the random Fourier feature method to approximate the arc-cosine kernels. Since the number of parameters scale exponentially with the number of layers, the authors further employ the polysketch to reduce the feature dimensionality. Theoretical analysis of the approximation error is also provided.

FInally, the authors demonstrate the efficacy of the proposed algorithms using extensive experiments on MNIST, CIFAR-10 and some UCI dataset. Experimental results shows that the proposed algorithm yields significant time savings with close to NTK prediction accuracy.

**Main Review:**

Clarity: The paper is clearly structured and well-written, though the authors are encouraged to carefully proof-read the manuscript to avoid some typos.

Originality: The nolvety of the paper is to combine the polynomial sketching with either Taylor expansion approximation or random Fourier feature approximation of the arc-cosine kernels. Both sketching and random Fourier feature approximation are well studied in literature in terms of their theoretical properties. This allows the authors from the current paper to provide detailed theoretical analysis of their proposed algorithms.

Quality \& Significance: This paper addresses an important issue of NTK learning by combining well studied methods in literature. In my opinion, the porposed algorithms are an important contribution to the literature.

Minor Comments:
Except the spectral approximation via leverage scores sampling, all the theoretical analyses focus on the kernel approximation error, which does not directly translate to the prediction accuracy. I am wondering if some analysis of the learning risk is possible.

**Time Spent Reviewing:**

15

---

> ### Author Response · Authors · 2021-08-10
> **Authors feedback**
>
> We thank the reviewer for the detailed and insightful comments. Regarding possible risk bound, the spectral approximation guarantee that we proved in Theorem 3 can directly provide statistical risk bounds for kernel ridge regression as shown in [Avron et al., 2017] (please see Lemma 2 therein). However our bounds on the point-wise distance between the kernel value and its approximation does not directly translate to guarantees on statistical risks.
>
> [Avron et al., 2017] Random Fourier features for kernel ridge regression: Approximation bounds and statistical guarantees, ICML, 2017

---

> > ### Comment · Reviewer_hkr8 · 2021-08-29
> > **Thanks for the response**
> >
> > Thank you for your response. I think the generalization property can be an interesting question to pursue further. I am happy to accept the paper.

---

### Decision · Program_Chairs · 2021-09-28

**Decision:**

Accept (Poster)

**Comment:**

This paper seems to give a substantially more practical method for using neural tangent kernels in practice, along with theoretical guarantees. Some issues of clarity and various small improvements to make to the paper came up in the discussion; make sure to incorporate those into the final version of the paper. I'd also like to encourage the authors to work towards getting their algorithm implemented in the https://github.com/google/neural-tangents package (if they haven't already), which would make practical uptake by users far more likely.

**Consistency Experiment:**

NeurIPS has a long history of experimentation. In 2014, NeurIPS ran an experiment in which 10% of submissions were reviewed by two independent committees to quantify the randomness in the review process. This year, we repeated a variant of this experiment to see how the quality of the review process has changed over time.  This paper was part of the experiment and was therefore assigned to two committees (consisting of reviewers, an Area Chair, and a Senior Area Chair) that reached independent decisions.  If both committees made the same recommendation, this recommendation was followed. If a single committee recommended acceptance, the paper was accepted (with the exception of a few cases in which the other committee identified what we considered a fatal flaw, e.g., an error in a key result).

Both committees reached the same decision: **Accept (Poster)**

The other committee assigned to the paper recommended **Accept (Poster)**.  You can find the other set of reviews, along with any follow up discussion with the authors here:
https://openreview.net/forum?id=vIRFiA658rh